# OpenReview forum: "Uncovering Intersectional Stereotypes in Humans and Large Language Models"
_ICLR.cc/2025/Conference — ICLR 2025 Conference Withdrawn Submission_

### Official Review · Reviewer_NoA5 · 2024-10-28

**Soundness:** 2
**Presentation:** 2
**Contribution:** 2
**Rating:** 3
**Confidence:** 3

**Summary:**

This paper's contribution is two-fold. First, it evaluates intersectional biases in LLMs by augmenting WinoBias using 25 demographic markers. Second, it conducts user study to uncover intersectional biases from humans.

**Strengths:**

The topic of intersectional biases in LLMs is an important topic.

**Weaknesses:**

W1: It’s unclear what exactly the authors measure as "fairness" in this study. They mention in line 154: “frame model unfairness as a disparity in model,” but there’s no clear quantification or metric provided in either Figure 1 or Section 2.3. Both sections would benefit from reporting a concrete fairness measure to improve interpretability.

W2: Section 2.3 claims that a decrease in “next-word probability for the referent occupation upon referent augmentation” signals bias. However, this comparison—between terms like “designer” and “black female designer”—seems unjustified as an indicator of intersectional bias. A more accurate measure of intersectional bias would compare (1) “black female designer” vs. “white female designer,” (2) “black female designer” vs. “white male designer,” and (3) “black female designer” vs. “black male designer.” The observed performance drop in the original comparison could simply reflect the general effect of adding more adjectives, or, as noted in line 251, “that current LLMs do not reason well about intersectional identities.”

W3: Table 3 shows that next-word probability decreases for both pro- and anti-stereotypical sentences, suggesting a general decline in co-reference resolution rather than a shift in bias. This raises questions about whether the metrics used here accurately represent bias.

W4: It’s challenging to discern any clear bias changes in Section 2.4 and Table 3. First, what exactly is measured as bias, is it |referent next-probability - 0.5|? If so, this number should be directly reported rather than the raw next-probability, as the latter obscures the intended bias signal.  Second, why bias is measured as |referent next-probability - 0.5| rather than |referent next-probability for the first occupation - referent next-probability for the second occupation|? The former measure could fluctuate due to factors unrelated to bias, such as other words consuming higher probabilities.

W5: While the authors wrote that “Figure 2 shows that human and LLM scores are highly correlated on the Pearson Correlation Coefficient”, I find human_C is correlated with both LLM_C and LLM_W. But human_W is not strongly correlated with either LLM_C and LLM_W. The authors should clarify these correlation patterns and discuss the implications to support their conclusions.

W6: I disagree with the authors’ claim in the abstract (and reiterated in line 77 and line 532) that their findings indicate a potential for LLMs to replace human subjects in social psychology research. The paper shows a correlation between human and LLM biases, but why would this correlation justifies that social psychology research could use LLM to replace human subjects and believe that the findings would still apply to the real human society? To substantiate this claim, the authors should establish clear criteria for when and how AI can be a substitute in social psychology research, demonstrate how their findings satisfy these criteria, and address the validity issues this substitution could introduce.

W7: As I noted in W6, the paper does not effectively address research question RQ2: 'Can LLMs help us detect social stereotypes that have not been studied before?' It lacks justifications to the conclusion that "the potential for LLMs to be used to conduct social psychology research that could otherwise be harmful to conduct with human subjects" beyond correlation.

W8: The paper first part on LLMs feels disconnected from the second part on human studies, making them more like two separate, less related studies.

W9: As the authors noted in the related works section, there is existing literature, including [1] and [2], that benchmarks and demonstrates intersectional biases in LLMs. In its current form, this paper’s results lack clarity regarding novel insights beyond what these studies have already shown. Simply demonstrating the presence of intersectional biases in LLMs doesn’t seem sufficient to warrant publication.

[1] Kirk, Hannah Rose, et al. "Bias out-of-the-box: An empirical analysis of intersectional occupational biases in popular generative language models." Advances in neural information processing systems 34 (2021): 2611-2624.
[2] Lalor, John P., et al. "Benchmarking intersectional biases in NLP." Proceedings of the 2022 conference of the North American chapter of the association for computational linguistics: Human language technologies. 2022.

**Questions:**

See weakness.

---

### Official Review · Reviewer_ihRB · 2024-10-29

**Soundness:** 1
**Presentation:** 1
**Contribution:** 1
**Rating:** 1
**Confidence:** 4

**Summary:**

The paper proposes an intersectional stereotype benchmark simply by adding words to an existing benchmark and jumps to the conclusion that intersectional stereotypical bias exists. In addition, it shows there exists a correlation between human annotators and large language models (LLMs), and claims LLMs could be used as a replacement for human subjects.

**Strengths:**

Sorry, but I can hardly tell a distinct advantage of this paper. I would reject this paper even if it's submitted to an NLP conference and I'm the reviewer there.

**Weaknesses:**

* There exists a huge discrepancy between the two research questions (how can you imagine combining these two topics, benchmark introduction and replacement of human subjects with LLMs, in one paper), and the so-called "two-fold contributions" seem to be the simple addition of two simple experiments and weak conclusions.
* The paper is poorly written. For example, they introduce a benchmark but never include benchmark statistics. Another instance is Figure 1, which is difficult to read. (Why not just select several representative words and show the trend?) Moreover, they list tables of numbers without informative captions to instruct the readers to interpret them. The paper writer seems to be naive. I believe when submitting to a conference, it's not the more incomprehensible tables/figures, the better. Every researcher conducts a lot of experiments to write a paper, and their disorganized compilation of all experiments conducted (even without significant conclusions) shows disrespect to peer reviewers.
* Their benchmark is simply adding words to a pre-existing benchmark. IMO could not be viewed as a contribution.
* Misuse of math terms. The uncertainty they mention should not be measured by testing with different samples but by the same sample with slightly changed prompts or models' random seeds. Otherwise, they cannot explain where the randomness comes from.
* Replacing humans with LLM is a straightforward idea but needs further consideration when applied to stereotype measurement. I believe the main reason why the stereotype measurement domain hesitates to replace human subjects with LLM is that LLM is trained on text, which implies if some social groups or perspectives are not fully represented with a large volume of LLM-trainable text, their voices are missing. I don't think the second "contribution" is significant or reasonable.
* I believe the author, to some extent, favors overstating their contributions. While they repeatedly claim that "to the best of their knowledge, they are the first to use uncertainty," I've seen many stereotype measurement works using the idea of statistics. Searching the keywords "stereotype statistical language model" on Google Scholar, pages of statistical stereotype measurement work pop up. Never include them in the related work section doesn't mean you are the first to propose the research idea.

**Questions:**

- How would you add transitions to the two research questions?
- What is your definition of uncertainty here? Where does the randomness come from?
- Please don't add that many disorganized figures and tables in the main body and appendix.
- Have you considered such an "under-represented" problem of some social groups when you propose to replace human subjects with LLMs?
- Did you conduct background research comprehensively to say "to the best of your knowledge, you are the first to use statistics ideas to measure stereotypes?"

---

### Official Review · Reviewer_rvkN · 2024-11-03

**Soundness:** 2
**Presentation:** 2
**Contribution:** 3
**Rating:** 5
**Confidence:** 3

**Summary:**

This paper tackles the pressing issue of intersectional bias in LLMs by introducing the WinoIdentity dataset, an extension of WinoBias that incorporates 25 demographic markers, allowing for a nuanced analysis of bias across intersecting identities. The authors propose uncertainty parity as a new fairness metric, aiming to capture bias by examining how the model’s uncertainty varies with demographic markers. Furthermore, the paper explores how stereotypes embedded in LLMs align with human stereotypes, using the warmth-competence model from social psychology. The findings highlight both the ethical risks and potential for LLMs to support social research on stereotypes without direct human involvement.

**Strengths:**

- Intersectional Focus: This paper advances the field by addressing intersectional biases in LLMs, a relatively unexplored area that is critical.
- WinoIdentity Dataset: The WinoIdentity benchmark is a well-thought-out dataset addition, facilitating research on intersectional biases and setting a foundation for future work in this area.
- New Fairness Metric: Uncertainty parity offers a fresh perspective on measuring fairness, emphasizing model confidence rather than just accuracy, which may lead to more nuanced interpretations of bias.
- Interdisciplinary Insight: Applying the warmth-competence model from social psychology bridges AI and human perception research, suggesting possible applications of LLMs in social psychology studies.

**Weaknesses:**

- Demographic Selection and Generalizability: The WinoIdentity dataset includes 25 demographic markers, but the criteria for selecting specific intersections of these markers are unclear. Clarifying whether the selections were guided by social science research or simply represent dataset limitations would strengthen understanding of the benchmark’s broader applicability.
- Justification of Uncertainty Parity: While uncertainty parity provides a novel way to assess bias, the choice to prioritize it over other fairness metrics (e.g., calibration, confidence intervals) is not fully explained. Understanding why this metric is more effective in detecting bias in intersectional contexts would help validate its utility.
- Stereotype Alignment Analysis: The paper reports statistically significant alignment between LLM stereotypes and human stereotypes but does not fully detail the statistical methods (e.g., effect sizes or confidence intervals) used to establish this alignment. Including these details would strengthen confidence in these findings.
- Lack of Interpretability in Refusal Mechanisms: While the paper suggests that LLMs exhibit greater uncertainty for marginalized identities, it does not clarify how the model decides to “refuse” certain responses based on low confidence. Insights into this boundary-setting process, especially in intersectional contexts, would make the model’s behavior more interpretable and applicable.

**Questions:**

- Could the authors elaborate on the criteria for selecting demographic intersections in WinoIdentity? Were these choices based on specific literature, practical dataset constraints, or observed model behaviors?
- Why was uncertainty parity selected as the primary metric for fairness? Could the authors compare this metric with more established metrics, and explain how it captures intersectional bias more effectively?
- How robust is the alignment between LLM and human stereotypes? Could the authors clarify the statistical methods used to validate this alignment, particularly regarding effect sizes and confidence levels?
- Are there mechanisms for users to interpret refusal behavior based on uncertainty? Given that marginalized identities see higher refusal rates, it would be helpful if users could understand when and why the model declines to answer, potentially through visual explanations or interpretability methods.

---

### Official Review · Reviewer_e3Fy · 2024-11-04

**Soundness:** 3
**Presentation:** 3
**Contribution:** 2
**Rating:** 3
**Confidence:** 3

**Summary:**

This paper investigates the biases of LLMs towards the intersections of different axes of identity. The authors derive a new fairness benchmark (WinoIdentity) for evaluating intersectional stereotypes in LLMs by augmenting WinoBias with 25 new demographic markers. The authors use their benchmark to evaluate five LLMs, finding that they exhibit uncertain performance on pronoun-occupation coreference resolution. The authors also use LLMs to detect stereotypes against intersectional identities, and find that LLM-generated stereotypes are aligned with human-generated stereotypes.

**Strengths:**

- The study covers five LLMs and 50 intersectional identities.

- The authors guard against robustness issues when prompting LLMs for bias by also considering diverse rephrasings of their prompts [1].

- The authors apply the methodology of [2] to consider disparities in the warmth and competence of social groups between gendered subgroups, as rated by humans and LLMs, extending previous findings about stereotypes.

[1] https://arxiv.org/abs/2210.04337

[2] https://psycnet.apa.org/record/2002-02942-002

**Weaknesses:**

- Blodgett et al. (2020) is about how bias is poorly conceptualized in NLP papers [1], not about how “LLMs learn and reproduce pre-existing biases in their training data” (lines 38-39). Furthermore, in relation to lines 43-47, Intersectionality is about interlocking mechanisms of social oppression, not just differential forms of discrimination [2].

- The authors’ claim that intersectional biases “have so far been overlooked in LLM fairness evaluations” is overly strong given that a handful of papers have considered this topic (e.g., [3, 4, 5]) and many papers have considered intersectional fairness issues in machine learning more broadly (see Table 5 in [2]).

- Some of the augmented prompts in Table 2 are unrealistic or unnatural. For example, in practice, hegemonic identities are often unmarked (see Table 2 in [6]). Furthermore, “at her, the Black woman” reads awkwardly. I also suggest changing “female pronouns” to “feminine pronouns” (and similarly,  “male pronouns” to “masculine pronouns”) in Section 2.1 to emphasize the distinctions between sex, gender, and pronouns [7].

- The authors should elaborate on their findings in Table 3. While it is clear that “pro-stereotypical bias does not transfer to intersectional identities” (line 221), are there any socially-grounded hypotheses for why this is the case? What would be needed to test these hypotheses? Are there any similarities among the identities for which there is not pro-stereotypical bias?

- It is unclear how the measurement of unfairness as “uncertainty” is novel. How does this differ from past works that have used log probabilities to, e.g., capture model uncertainty about pronouns used by an individual [8].

- While the authors do acknowledge this in Section 4, various studies have argued against the use of LLMs as replacements for humans in social science research, on the grounds of this practice not fostering inclusion, flattening representations of marginalized communities, and essentializing identities [9, 10]. This is supported by the authors’ finding that “LLMs exacerbate negative stereotypes” (line 377).

- A large part of the authors’ methodology appears to be derived from [11].

[1] https://aclanthology.org/2020.acl-main.485/

[2] https://dl.acm.org/doi/10.1145/3600211.3604705

[3] https://aclanthology.org/2023.findings-emnlp.575/

[4] https://ojs.aaai.org/index.php/AIES/article/view/31748/33915

[5] https://aclanthology.org/2024.gebnlp-1.3/

[6] https://aclanthology.org/2021.acl-long.81/

[7] https://aclanthology.org/2021.emnlp-main.150/

[8] https://aclanthology.org/2023.acl-long.293/

[9] https://dl.acm.org/doi/full/10.1145/3613904.3642703

[10] https://arxiv.org/abs/2402.01908

[11] https://psycnet.apa.org/record/2002-02942-002

**Questions:**

- Lines 147-149: Could the authors elaborate on how the proposed benchmark measures stereotypes in particular? It seems that there is a gap between measuring explicit stereotypes (i.e., harmful generalizations about groups generated by LLMs) vs. evaluating if LLMs are not “distracted” by inclusion of identity terms when performing coreference resolution.

- Lines 202-203: The authors interpret “next-token probability close to 0.5” as indicative of the model being “uncertainly correct.” Does this interpretation of the next-token probabilities as a measure of certainty assume that the probabilities are well-calibrated?

- Where do the authors measure the “statistically significant” (line 28) alignment on stereotypes between humans and LLMs?

**Details Of Ethics Concerns:**

The authors ask humans to rate societal perceptions of the warmth and competence of social groups. It is not indicated whether the human study was reviewed by a research review board.

---

### Note · Authors · 2024-11-20

**Comment:**

Withdrawing this paper from consideration at ICLR. We believe that this work is a valuable contribution to the community and will work to communicate this more clearly while strengthening the work as a result of feedback from most of the reviewers.

**Withdrawal Confirmation:**

I have read and agree with the venue's withdrawal policy on behalf of myself and my co-authors.